# Synthesis and Characterisation of Novel Temperature and pH Sensitive Physically Cross-Linked Poly(N-vinylcaprolactam-co-itaconic Acid) Hydrogels for Drug Delivery

**DOI:** 10.3390/gels5030041

**Published:** 2019-08-29

**Authors:** Megan Fallon, Shane Halligan, Romina Pezzoli, Luke Geever, Clement Higginbotham

**Affiliations:** 1Materials Research Institute, Athlone Institute of Technology, Dublin Road, Athlone, Co. Westmeath N37 F6D7, Ireland; 2Applied Polymer Technologies Gateway, Materials Research Institute, Athlone Institute of Technology, Dublin Road, Athlone, Co. Westmeath N37 HD68, Ireland

**Keywords:** Poly(N-vinylcaprolactam), itaconic acid, temperature and pH responsive, physically cross-linked hydrogel, drug delivery

## Abstract

Previous studies involving poly N-vinylcaprolactam (PNVCL) and itaconic acid (IA) have synthesised the hydrogels with the presence of a solvent and a crosslinker, producing chemically crosslinked hydrogel systems. In this study, however, temperature sensitive PNVCL was physically crosslinked with a pH-sensitive comonomer IA through ultraviolet (UV) free-radical polymerization, without the presence of a solvent, to produce hydrogels with dual sensitivity. The attenuated total reflectance Fourier transform infrared (ATR-FTIR) spectroscopy indicated successful polymerisation of the hydrogels. The temperature and pH sensitivity of the hydrogels was investigated. The lower critical solution temperature (LCST) of the gels was determined using the UV spectrometry and it was found that the incorporation of IA decreased the LCST. Rheology was conducted to investigate the mechanical and viscoelastic properties of the hydrogels, with results indicating IA that enhances the mechanical properties of the gels. Swelling studies were carried out at ~20 °C and 37 °C in different buffer solutions simulating the gastrointestinal tract (pH 2.2 and pH 6.8). In acidic conditions, the gels showed gradual increase in swelling while remaining structurally intact. While in basic conditions, the gels had a burst in swelling and began to gradually degrade after 30 min. Results were similar for drug release studies. Acetaminophen was incorporated into the hydrogels. Drug dissolution studies were carried out at 37 °C in pH 2.2 and pH 6.8. It was found that <20% of acetaminophen was released from the gels in pH 2.2, whereas the maximum drug released at pH 6.8 was 74%. Cytotoxicity studies also demonstrated the hydrogels to be highly biocompatible. These results indicate that physically crosslinked P(NVCL-IA) gels possess dual pH and temperature sensitive properties, which may be beneficial for biomedical applications such as drug delivery.

## 1. Introduction

Hydrogels that can respond to a number of different environmental factors are commonly referred to as stimuli sensitive hydrogels. Stimuli sensitive polymers are an attractive option for the development of “smart” delivery devices for a wide number of active pharmaceutical ingredients (API) capable of synchronizing their release according to changes in the metabolic states of the body [1]. Among all the different environmental factors, the pH and temperature are the most widely used. Copolymers with the presence of a pH sensitive monomer together with a temperature sensitive monomer allow for the hydrogel to undergo volume changes in response to the pH and temperature [2]. This results in a tailored release of the hydrogel to meet specific process or system needs which require the temperature and pH as a factor, such as anti-cancer treatments [3].

The fabrication of temperature and pH sensitive hydrogels have become attractive in the last decade. Particular attention has been paid to reversible or physical gels in which the cross-linking occurs by molecular entanglements and/or secondary forces such as H-bonding and hydrophobic forces. Physically cross-linked gels are of greater interest in modern biomedical applications due to the ability of the gels to degrade and eventually disintegrate and dissolve over time. This allows for a less invasive treatment for the patient as the medical device does not need to be removed after use [4]. 

Poly(N-vinylcaprolactam) (PNVCL) is a thermosensitive polymer which has gained major interest with researchers due to its attractive characteristics such as biocompatibility, solubility, and thermosensitivity. In addition, the lower critical solution temperature (LCST) of PNVCL (32–34 °C) is near the range of the physiological temperature [5]. The LCST behaviour of PNVCL is sensitive to alterations in the polymer concentration, the molecular weight of the polymer and the composition of the solution, allowing for the ability to control the temperature sensitivity of PNVCL. This unique quality makes this polymer a desirable material for biomedical applications [6]. 

Itaconic acid (IA) is an acid-based monomer that offers a number of advantages to other carboxylic acid monomers; one such advantage is that IA has two ionisable groups with different pKa values (3.85 and 5.44), which allows for increased pH sensitivity in the system. In acidic pHs (below its pKa values), IA remains stable and swelling is minimal. At pHs higher than the pKa of IA, due to the ionisation of carboxylic groups, the gels are swollen and complexes begin to disintegrate. This makes IA a very attractive material for oral drug delivery as in acidic conditions the drug is protected. As the device passes through the gastrointestinal (GI) tract and the pH increases, the gel swells and drug is released, allowing for a targeted controlled drug delivery [7,8]. Most recently, IA was used as a copolymer to achieve pH sensitivity. Spasojević et al. (2015) found that by adding IA to a Poly(N-isopropyl acrylamide) (PNiPAAm) network, it allowed the hydrogel swelling and the phase transition of P(NiPAAm/IA) to alter the properties of the hydrogel [7].

Cavus and Cakal (2012) synthesised NVCL-IA in ethanol by using the free radical cross-linking polymerisation method at 60 °C for 24 h in the presence of azobis(isobutyronitrile) (AIBN) as the initiator and allyl methacrylate (AMA) as the cross-linking agent, giving rise to a chemically crosslinked hydrogel system. The swelling, diffusion, temperature and pH properties of the synthesised hydrogels were examined. The main aim of that study was to investigate the effect that the synthesis medium had on the properties of the gels, by using ethanol and ethanol/water mixture as the synthesis medium. It was found that the synthesis medium considerably affected the swelling behaviour and the percentage of gelation of the gels [9]. Other studies have developed chemically crosslinked PNVCL-IA based hydrogel systems, however, no studies have investigated the properties of physically crosslinked PNVCL-IA based hydrogels synthesised without the presence of a solvent [10,11]. 

The aim of this study was to synthesise novel physically crosslinked P(NVCL-IA) hydrogels via free radical polymerization without the presence of any solvent. In addition, to investigate the influence of the pH-sensitive IA monomer on the mechanical, thermal, chemical and biocompatible characteristics of PNVCL and to evaluate the potential use of dual sensitive P(NVCL-IA) hydrogels as controlled drug delivery systems. 

## 2. Results and Discussion

### 2.1. Preparation of Samples

Physically crosslinked PNVCL based hydrogels were synthesised via free-radical photopolymerisation based on methods used by Halligan et al. (2017) [12]. Before use, NVCL was heated in hot water bath at 60 °C for 1 h. Without the presence of any solvent, various concentrations of the IA monomer and 1.0 wt% of the photoinitiator was dissolved in NVCL at different concentrations, see Table 1.

Homogeneous solutions of P(NVCL-IA) gels were prepared by dissolving appropriate amounts of the hydrogels (3 wt%) in distilled water. These samples were only used for the purpose of determining the LCST of the hydrogels. 

### 2.2. Characterisation Methods

#### 2.2.1. Attenuated Total Reflectance Fourier Transform Infrared Spectroscopy

To confirm photopolymerisation, the attenuated total reflectance Fourier transform infrared spectroscopy (ATR-FTIR) analysis was carried out on P(NVCL-co-IA) samples. Prior to the analysis, all samples were dried in a vacuum oven at 40 °C for 24 h. To ensure that photopolymerisation occurred, the peaks associated with the NVCL monomer should disappear in the synthesised hydrogel spectra. The C=C band can be found in the NVCL monomer spectrum at 1655 cm^−1^, however this band disappears in the synthesised hydrogel spectra with only a single band found between 1620 and 1631 cm^−1^ due to the amide C=O stretching. Additionally, the disappearance of the band found at 985 cm^−1^, due to the =CH_2_ bending, is an indication of successful polymerisation as it confirms the opening of the olefinic double bond. This is evident in the spectra of the synthesised hydrogels (as shown in Figure 1). These findings correspond with other studies that confirm that the transformation of the monomers to the polymer was successful for PNVCL [6,13,14]. 

The P(NVCL-IA) hydrogel spectra exhibited new peaks not typically associated with PNVCL, which indicates that the copolymer reaction successfully took place as represented in Figure 1. One such peak occurred at around 1719 cm^−1^ which corresponds to the C=O stretching of the carboxyl group of IA [15,16]. The C=O of the carboxyl group increased in intensity as the concentration of IA increased in the samples. The C=O peak of the carboxyl group displayed a shift when polymerisation occurred. In the IA monomer, the C=O peak was found at 1691 cm^−1^, whereas for the copolymer gels this band shifted to 1716 and 1719 cm^−1^. Similar results were found in literature, the bands between 1716 and 1719 cm^−1^ were attributed to the C=O stretching of the carboxyl group in IA [9].

#### 2.2.2. Differential Scanning Calorimetry

The differential scanning calorimetry (DSC) analysis was carried out to determine the effect that the incorporation of IA had on the thermal properties of P(NVCL-co-IA) hydrogels. One broad transition was observed between 80–180 °C for all the PNVCL based hydrogels, as shown in Figure 2. This transition was identified as the glass transition temperature (T_g_). The T_g_ of a hydrogel is an important parameter to consider in the development of a controlled drug delivery system. Polymer-drug delivery systems below their T_g_ value result in small diffusion rates, while above their T_g_ the polymer chains increase in mobility, resulting in greater diffusion rates [17]. In the case of storage, a controlled drug delivery system with a T_g_ greater than room temperature is an advantage. It is widely reported in literature that PNVCL was determined to have a T_g_ of around 145 °C with Lebedev et al. reporting a T_g_ of 147 °C for PNVCL [18].

For the PNVCL homopolymer synthesised in this study, the T_g_ was determined to be approximately 166 °C. The difference in T_g_ compared to the literature can be associated with various factors such as molecular weight, molecular weight distribution and purity. The T_g_ of the PNVCL based copolymers was found to be dependent on the concentration of IA. Increasing the concentration of IA led to a decrease in the glass transition temperature value. This decrease is likely due to the IA low molecular weight (130.10 g/mol) allowing for greater flexibility in the polymers’ chains resulting in lowered T_g_ [19]. Throughout this work, a single distinctive endotherm is exhibited for the analysed samples representing the T_g_, thus the samples can be characterised as random NVCL-IA copolymers.

#### 2.2.3. Phase Transition Determination 

Phase transition temperature is an important parameter determining the potential use of hydrogels in drug delivery [20]. In this study, phase transition temperature was investigated to evaluate the effects of the incorporation of IA on the LCST of PNVCL-IA hydrogels. UV–spectroscopy was carried out on the PNVCL and PNVCL-IA aqueous samples to determine the phase transition temperature by monitoring the change in optical transmittance at 500 nm over the temperature range 20–44 °C. The LCST of the aqueous solutions was determined at the temperature which showed an optical transmittance of 50%. The PNVCL phase transition temperature can be tailored by altering the ratio of NVCL to IA. As a result of this alteration the LCST is affected. This is due to the water content increasing the hydrogen bonding interactions between water and the polymer, which requires more thermal energy to break the water structure, resulting in an increase in the LCST.

This study modified the phase transition of PNVCL by changing the concentration of IA. PNVCL is affected by the IA concentration as it can change in the polymer-solvent interaction. In literature, it is reported that PNVCL was found to have a LCST of about 32 °C [20]. In this study, the PNVCL homopolymer was found to have an LCST of 31.8 °C. Increases in the IA concentration leads to a decrease in the phase transition temperature, which can be seen in Figure 3 and Table 2. 

The LCST of P(NVCL 95-IA 5) was determined to be 29.8 °C compared to P(NVCL 90-IA 10) with a LCST of 28.4 °C. This is possibly due to the high content of the carboxyl group in IA, which affects the phase transition of P(NVCL-IA). The LCST of the copolymers were determined in water (pH 7), which is above the pKa of the ionisable groups. This may result in repulsion between the −COO¯ groups from IA, which leads to more hydrophobic interactions and increases the overall effective hydrophobicity of the copolymers, thus lowering the phase transition temperature. 

This decrease in phase transition is due to the solution behaviour of the PNVCL and corresponds to a Type 1 Flory-Huggins demixing behaviour with phase transition [21]. Since the LCST of the P(NVCL-IA) hydrogels is lower than the average body temperature, the gels will form a stable shell in the physiological environment that can be used for controlled drug delivery [12,22].

#### 2.2.4. Swelling Studies

To examine the pH and temperature sensitivity of the P(NVCL-IA) hydrogels, swelling studies were carried out on samples in two different buffers simulating the gastrointestinal track (pH 2.2 and pH 6.8) at room temperature (~20 °C) and at 37 °C. If polymerisation was successful, P(NVCL-IA) hydrogels should have unique swelling profiles in response to different pH buffers and temperatures. The swelling profiles of hydrogels with dual responsivity can be tuned to a specific pH and temperature, which plays an essential role in determining their use in biomedical applications such as drug delivery and tissue engineering [23].

The pH sensitivity of the samples was determined by swelling the gels in different buffer solutions at room temperature (~20 °C). Samples swollen in pH 6.8 reached their maximum swelling capacity after 30 min and after this the gels began to break down. Increasing the concentration of IA, increased the swelling of all the samples in buffer pH 6.8. As seen in Figure 4, samples in pH 6.8 at ~20 °C tend to dissolve rapidly, and after 300 min, the polymers were completely dissolved. Swelling studies carried out in a buffer of pH 2.2 showed that the samples swelled but did not break down. The samples showed physically crosslinked characteristics, where the molecular entanglements and/or secondary forces such as ionic, H-bonding or hydrophobic forces play the main role in forming the network. It is possible to dissolve physically crosslinked gels by changing environmental conditions, such as pH, and the ionic strength of solution or temperature [23]. The presence of hydrogen bonding between the PNVCL and IA in the hydrogel depends on the pH of the medium causing a difference in the degree of swelling and dissolution. At pH 2.2, which is below the pKa of the samples, the carboxylic groups in the polymer are in a non-ionised state and the samples swelled without dissolving. At pH 6.8 carboxylic acids are in the dissociated form (COO¯). The repulsions between the negative charges lead to a chain stretch, thus resulting in the samples dissolving quickly [14].

The incorporation of hydrophobic or hydrophilic monomers introduces new properties to PNVCL-based hydrogels while also retaining their temperature sensitivity [9]. In this study, the effect of temperature on the swelling behaviour of PNVCL-based hydrogels was investigated by swelling samples in different buffer solutions below (~20 °C) and above (37 °C) the LCST of the gels. Below LCST, samples with 100% PNVCL swelled less compared to the samples containing IA. This decrease in swelling is due to the hydrogen bonding between the polymer segments and water, which eventually leads to polymer dissolution [14,24]. Above LCST, hydrogen bonding is weakened and hydrophobic segments become dominant, leading to precipitation of the polymer [12,24].

#### 2.2.5. Rheological Measurements

The mechanical properties and viscoelastic behaviour of the hydrogels were investigated using frequency sweep rheological measurements. The types of interaction, such as covalent bonding, hydrogen bonding or molecular entanglements, within a hydrogel network can influence the mechanical properties [25]. Results found that the viscous region dominates over the elastic region as the shear loss modulus (G”) was greater than that of the storage modulus (G’). This may be due to weak bonds present within the hydrogel matrix. Prior to testing, samples were hydrated in a PBS buffer at room temperature. At these conditions, the hydrogel system becomes more hydrophilic and dissolution may begin, thus weakening the hydrogel matrix and hence producing a less elastic and a more viscous system [25]. 

Figure 5 shows that the storage modulus (G’) of the samples increases with the content of IA. The incorporation of IA leads to more interactions within the hydrogel matrix, thus enhancing the mechanical properties of the gels [25,26]. For oral drug delivery systems, increasing the mechanical properties of a viscous material is beneficial for protecting the drug throughout the gastrointestinal tract. 

### 2.3. Drug Release Studies

#### Drug Dissolution

In recent years, enormous advances have occurred in polymer-based controlled-release drug delivery systems. An ideal drug delivery system should respond to physiological changes and alter the release of drug accordingly as a result. Hydrogels can be manufactured to exhibit changes in their swelling behaviour, network structure, permeability, or mechanical strength in response to different stimuli such as pH and temperature. Stimuli-responsive hydrogels allow for desired drug release as required in response to physiological need [27]. Controlled drug release reduces administration times and undesired side effects, while also improving patients’ compliance and comfort [28]. 

In this study, acetaminophen was chosen as a model drug. The drug (1.0 wt%) was encapsulated into the hydrogels as prepared in Section 4.2. The aim of dissolution studies was to determine the pH-sensitive drug release properties of P(NVCL-IA) hydrogels and the effect of the IA content on the drug release. To assess the pH-sensitivity of the hydrogels in-vitro drug release studies were carried out in gastric (pH 2.2) and intestinal (pH 6.8) pH conditions at 37 °C. Figure 6 shows the release profiles of acetaminophen from the samples at different pHs.

From these results it is evident that pH influenced the rate and amount of drug released from the hydrogels. In pH 6.8, all samples containing IA exhibited an increase in the drug release compared to samples tested in pH 2.2. This may be due to the higher degree of swelling due to the ionisation of carboxylic groups of IA in the hydrogel networks at pH 6.8 [6,29]. It was also found that the samples with the higher concentration of IA showed greater drug release in pH 6.8. P(NVCL 90-IA 10) exhibited a total of 74% release of drug after 6 h compared to P(NVCL 100) which only released 14%. This is possibly due to the hydrophilicity of the hydrogel matrix being increased by the incorporation of IA. 

In pH 2.2, the hydrogels containing IA showed less drug release compared to P(NVCL 100). This may be due to the low swelling capacity of the gels containing IA at this pH. As a potential drug delivery system, these P(NVCL-IA) hydrogels would be suited to deliver drugs to areas with a basic pH such as the intestine (pH 6.8). These gels have the capability to protect a drug in the acidic conditions of the stomach and release drug in response to the change in the physiological pH.

One study loaded P(NVCL-IA) based hydrogels with Nadolol and investigated the drug release behaviour of the microgels. It was found that after 48 h only 48% of Nadolol was released at pH 7.2 at 37 °C. This may be due to the hydrogels being chemically crosslinked and Nadolol having a high electrostatic interaction with the hydrogel network [30]. For oral drug delivery systems, physically crosslinked hydrogels may be more beneficial as they have the capability to release all the drug before completely passing though the gastrointestinal tract [31].

### 2.4. Cytoxicity Studies

Cytotoxic effects of three different concentrations (0.2, 0.6, 1.0 mg/mL) of each formulation against cultured Hep G2 cells were determined using the MTT assay. Results displayed in Figure 7 show that all formulations at every concentration are very biocompatible. This is similar to studies reported in literature which state that PNVCL-based hydrogels to be highly biocompatible [32,33]. In this study, the cell viability at 0.2 mg/mL is greater than or almost equal to the untreated control group, indicating excellent biocompatibility. Although, the cell viability decreases as the concentration of the hydrogel samples increase, results are still relatively high with cell viability varying from 109% to 80% at 1.0 mg/mL. Medeiros et al. synthesised chemically crosslinked P(NVCL-IA) microgels by precipitation polymerization [34]. The cytotoxicity of the gels were evaluated in-vitro by LDH leakage. Results showed that cytotoxicity increased as sample concentration increased. This may be due to the presence of the crosslinker in the formulations. This is one disadvantage of using chemically crosslinked hydrogels for biomedical applications as crosslinkers can induce cytotoxicity [34]. 

Overall, these results provide evidence that these hydrogels produce no serious toxic side effects to Hep G2 cell lines and thereby can be used as potential drug delivery systems. 

## 3. Conclusions

Physically crosslinked polymers of PNVCL samples were prepared using free radical polymerization using UV light. PNVCL based samples were combined with a pH sensitive monomer in an attempt to make dual responsive hydrogels. To ensure that polymerisation had occurred, the ATR-FTIR and DSC analysis was carried out and found that polymerisation was successful. The LCST was determined by the UV spectrometry where a decrease in the phase transition temperature was found. This decrease is most likely due to an increase in the IA content. The incorporation of IA also enhanced the mechanical properties which was found out by carrying out rheological measurements. The swelling analysis was conducted to examine the pH and temperature sensitivity of the hydrogels. The pH sensitivity was examined by the swelling samples in buffer solutions of different pH values (pH 2.2 and pH 6.8). Temperature sensitivity was examined by swelling the samples at temperatures below and above the determined LCST of the gels (~20 °C and 37 °C). Drug release studies were carried out to establish the pH sensitivity of the hydrogels. The incorporation of IA allowed for an increase of the drug released at pH 6.8. While in pH 2.2, small amounts of the drug were released. P(NVCL-IA) hydrogels may have the potential to protect a drug in the acidic conditions of the stomach and release a drug in response to changes in the physiological pH, making these materials very attractive for targeted drug delivery. Finally, the MTT assay revealed the hydrogels to be highly biocompatible and no evidence of toxic side effects. Overall, these results indicate these hydrogels have potential use as an oral drug delivery system. 

## 4. Materials and Methods

### 4.1. Materials

Monomers N-Vinylcaprolactam (NVCL) (98%) and itaconic acid (IA) were obtained from Sigma Aldrich (Arklow, Ireland). NVCL, which has a molecular weight of 139.19 g/mol, was stored at a temperature between 2 and 8 °C while IA, with a molecular weight of 130.10 g/mol, was stored at room temperature. 

1-[-4-(2-Hydroxyethoxy)-phenyl]-2-hydroxy-2-methyl-1-propan-1-one (Irgacure® 2959) was obtained from Ciba specialty chemicals (Basel, Switzerland). 

For the preparation of the buffer solutions, potassium biphthalate and hydrochloric acid were obtained from Sigma Aldrich. Potassium chloride was supplied from Merck (Darmstadt, Germany). For the preparation of the mobile phase, methanol and glacial acetic acid were both supplied by Sigma Aldrich. Acetaminophen which was incorporated into the gels, as a model drug, was supplied by Sigma Aldrich. All materials were used as received.

### 4.2. Hydrogel Synthesis

The polymers investigated in this study were prepared by free-radical polymerisation using ultraviolet (UV) light. These polymers were synthesised via physical crosslinking using a UV curing system (Dr. Gröbel UV-Elektronik GmbH) (Ettlingen, Germany). This particular irradiation chamber is a controlled radiation source with 20 UV-tubes that provide a spectral range between 315–400 nm at an average intensity of 10–13.5 mW/cm^2^. The monomer mixtures were prepared by combining desired amounts of the monomers and 1.0 wt% photoinitiator. The composition of the materials is listed in Table 1. The batches were placed in a 50 mL beaker and mixed using a magnetic stirrer for 20 min until a homogeneous mixture was achieved. The solutions were pipetted into silicone moulds that contained disc impressions (23 mm diameter, 2.2 mm thickness). Photopolymerisation was carried out at 15 min intervals where the samples were turned to ensure gels got the same level of UV exposure during the polymerisation process.

### 4.3. Preparation of Aqueous Solutions of Hydrogels

Homogeneous solutions of each sample were prepared by weighing appropriate amounts of the hydrogel (3 wt%) and distilled water. The mixtures were left at room temperature until completely dissolved. These aqueous solutions were produced for subsequent use in lower critical solution temperature (LCST) measurements.

### 4.4. Characterisation Methods

#### 4.4.1. Attenuated Total Reflectance Fourier Transform Infrared Spectroscopy 

The ATR-FTIR spectroscopy was carried out on a Perkin Elmer Spectrum One spectrometer, fitted with a universal ATR sampling accessory. All data was recorded at room temperature (<20 °C), in the spectral range of 4000–650 cm^−1^, utilising a 4 scan per sample cycle and a fixed universal compression force of 80 N. Subsequent analysis was performed using the Spectrum software.

#### 4.4.2. Differential Scanning Calorimetry

DSC studies were performed on all samples (TA Instrument DSC 2920 Modulated DSC). The DSC was calibrated with indium standards. The samples had a dry weight between 8 mg to 12 mg. Aluminium pans were used to contain the samples for the DSC experiment. All samples were examined under a pure nitrogen atmosphere. During testing, all samples were ramped from 20 °C to 200 °C at a rate of 10 °C/min. The analysis was conducted on all samples and the glass transition temperature (T_g_) was recorded.

#### 4.4.3. Phase Transition Determination 

The LCST behaviour of the samples was also investigated using a UV-spectroscopy synergy HT BioTek plate reader (Dublin, Ireland). The transmittance of the samples in aqueous polymer solutions was measured at 500 nm at intervals of 1 °C in the range of 22–44 °C. The solution was allowed to equilibrate at each temperature for 20 min before the transmittance was measured.

#### 4.4.4. Swelling Studies

After photopolymerisation, the samples were placed in a vacuum oven for 24 h at 40 °C to remove any moisture. Once dried, the apparent dry weights were measured. The swelling behaviour of the samples were tested in triplicate in different buffer solutions at room temperature (~20°C) and at 40 °C. Hydrochloric acid buffer (pH 2.2) and phosphate buffer (pH 6.8) were prepared in accordance to USP standards. The pH of the buffer solutions was confirmed by a Jenway pH meter, model 3520, which was calibrated by standard buffer solutions. Each sample was placed in a petri dish containing the buffer solution. At predetermined time intervals, the polymer was removed and blotted with filter paper to remove any excess water. The wet sample weight was then recorded. The samples were re-submerged into the fresh buffer solution and the procedure was repeated until the polymer was completely dissolved. The polymers swelling ratio was calculated using Equation (1).
Swelling Ratio (%) = (Wt/Wo) × 100(1)
where Wt is the weight of the gel at a predetermined swelling time and Wo is the dry mass weight of the gels.

#### 4.4.5. Rheological Measurements

Rheological measurements were carried out using the TA Discovery Series Parallel Plate Rheometer fitted with a Peltier temperature control. Samples were allowed to swell in the PBS buffer (pH 7.4) until an equilibrium swollen state was achieved. Prior to testing, the samples were blotted free of water to avoid slippage. The tests were performed at 37 °C ± 0.1 using a 20 mm stainless steel parallel plate. A compression load of 3 N ± 0.5 was exerted on the hydrogels in all cases. Experiments were carried out at a constant strain of 1%, verified by an amplitude sweep, with frequency ranging from 1.0 × 10^0^ to 1.0 × 10^3^ rads/s.

### 4.5. Drug Release Studies

#### 4.5.1. Drug Dissolution

The hydrogels were prepared as per Section 2.2 with 1.0 wt% of acetaminophen incorporated into the gels. Dissolution tests were carried out on the hydrogels using a Distek dissolution system according to the USP Dissolution apparatus 2. The hydrogels were tested in triplicate in buffer solutions of pH 6.8 and pH 2.2 simulating gastrointestinal tract conditions. All tests were carried out at a controlled temperature of 37 °C to represent body temperature. The stir rate was set at 50 rpm with 900 mL of buffer solution in each vessel. Samples were withdrawn at predetermined time intervals, every 30 min for the first 3 h and a final sample was withdrawn after 6 h. Samples were filtered using 0.45 µm filters. The drug release profile was determined using a high performance liquid chromatography (HPLC).

#### 4.5.2. High Performance Liquid Chromatography

The HPLC system consisting of a Waters 1515 isocratic pump combined with Waters 717 plus autosampler and Waters 2487 dual λ absorbance detector was used in this study. A 150 mm × 4.6 mm Thermo Scientific ODS Hypersil column with a particle size of 5 µm was used for the separation and quantification of acetaminophen in the release medium. The mobile phase was prepared according to the USP method by using the HPLC grade methanol, water and glacial acetic acid in the ratio of 69:28:3. A flow rate of 2 mL/min was maintained during the procedure and the detector was set at 275 nm. The calibration graph for acetaminophen was obtained by plotting the peak area versus concentration and the corresponding regression equation was used to calculate the concentration of the unknown. The dissolution profile was observed from a plot of time versus % Release.

### 4.6. Cytotoxicity Studies 

Cytotoxicity of the polymerized hydrogels were assessed using the MTT assay. Hep G2 cells were cultured in Dubecco’s modified eagle medium (DMEM) supplemented with 10% fetal bovine serum, penicillin/streptomycin and L-glutamine at 37 °C with 5 % CO_2_. For toxicity testing, three different concentrations (0.2, 0.6, 1.0 mg/mL) of each hydrogel formulation were dissolved in media. Hep G2 cells were seeded at 1 × 10^4^ cells per 100 µL of media in 96-well plates and incubated in an atmosphere of 5 % CO_2_ at 37 °C. Once the desired confluency was obtained, the cells were washed with PBS buffer and treated with 100 µL of the different hydrogel concentrations for 24 h at 37 °C with 5 % CO_2_ atmosphere. A 0.5 mg/mL MTT solution was prepared by dissolving 50 mg of MTT salt in 10 mL of sterile PBS buffer and then filter sterilized. This solution was further diluted in media (1:10 dilution) and used for the assay. Once the test media was removed, cells were incubated with the 0.5 mg/mL MTT solution for 4 h to form formazan crystals. After this, the solution was removed and 100 µL of di methyl sulfoxide (DMSO) was added and the resulting solution was measured for absorbance at 570 nm using a microplate reader. Results are presented as percentage cell viability. 

## Figures and Tables

**Figure 1 gels-05-00041-f001:**
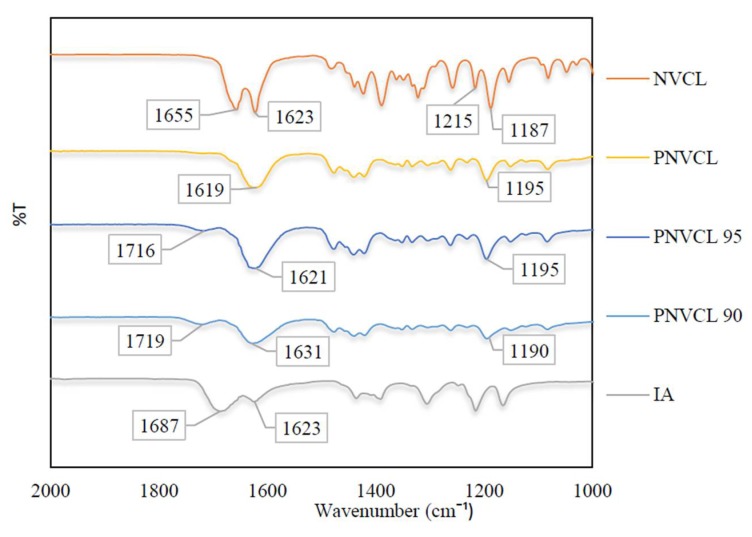
ATR-FTIR spectra of NVCL, IA and PNVCL based hydrogels from wavelength 2000–1000 cm^−1^.

**Figure 2 gels-05-00041-f002:**
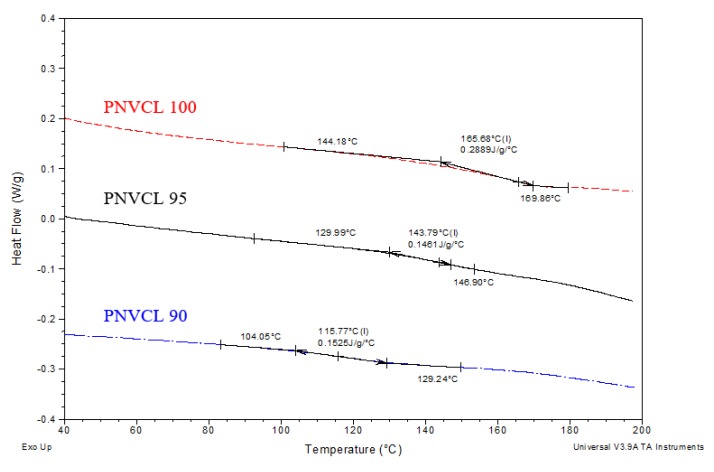
DSC thermograph of PNVCL based hydrogels.

**Figure 3 gels-05-00041-f003:**
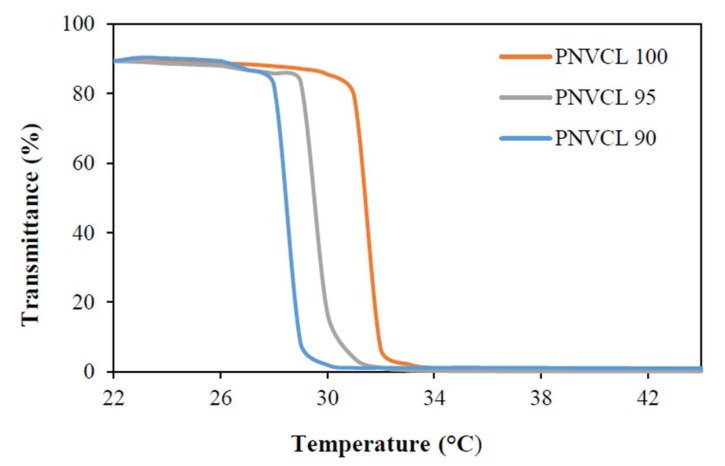
UV-spectrometry illustrating the phase transition of PNVCL based hydrogels.

**Figure 4 gels-05-00041-f004:**
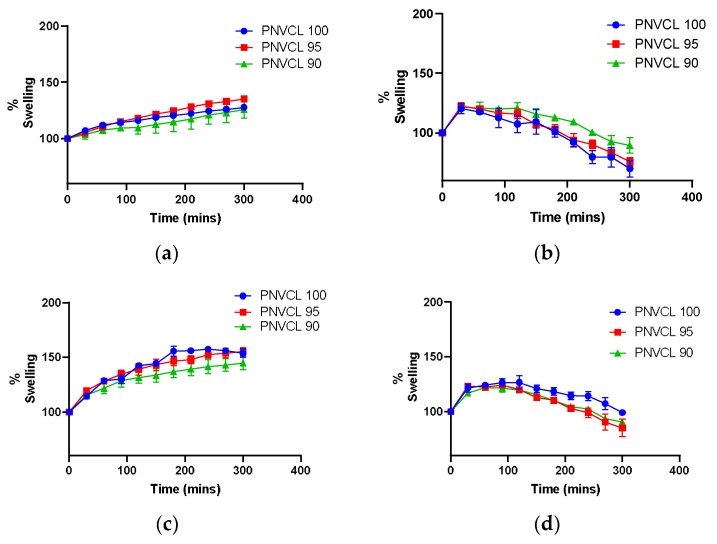
Swelling profiles of PNVCL-based hydrogels in: (**a**) pH 2.2 at ~20 °C; (**b**) pH 6.8 at ~20 °C; (**c**) pH 2.2 at 37 °C; (**d**) pH 6.8 at 37 °C.

**Figure 5 gels-05-00041-f005:**
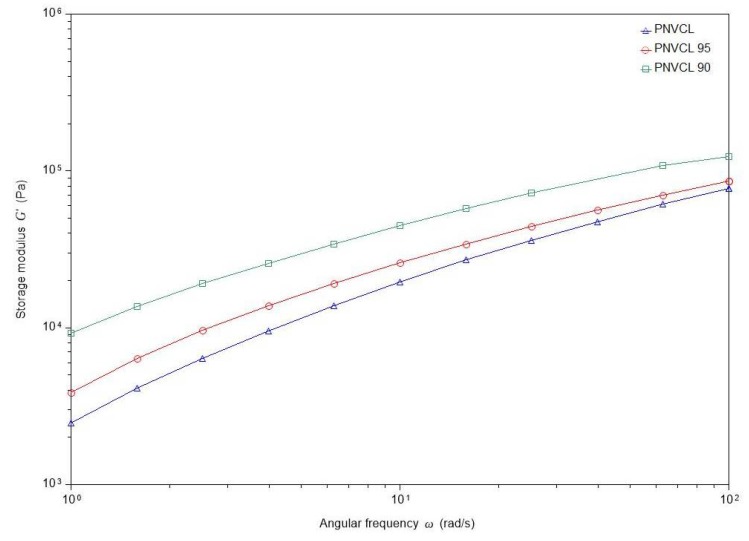
Frequency sweep results for hydrogel samples.

**Figure 6 gels-05-00041-f006:**
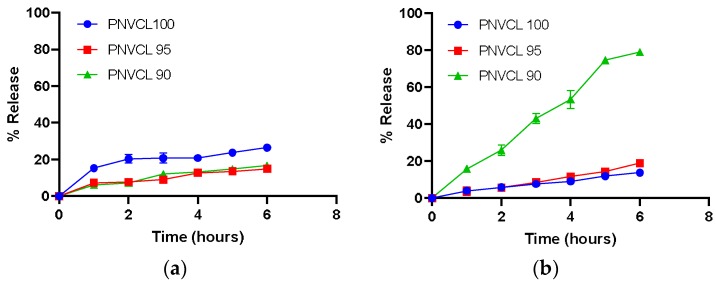
Release profile of acetaminophen from PNVCL-based hydrogels at: (**a**) pH 2.2; and (**b**) pH 6.8.

**Figure 7 gels-05-00041-f007:**
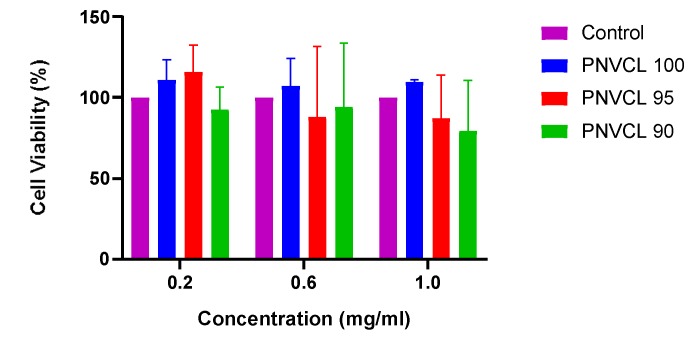
% Cell viability, using the MTT assay, of Hep G2 cells treated with different concentrations of hydrogel samples for 24 hours.

**Table 1 gels-05-00041-t001:** Physically crosslinked Poly(N-vinylcaprolactam) (PNVCL) based hydrogels containing 1.0 wt% of the photoinitiator Irgacure® 2959 and 5/10 wt% of itaconic acid (IA).

Polymer Code	NVCL (wt%)	IA (wt%)	Irgacure® 2959
PNVCL 100	100	0	1.0
PNVCL 95-IA 5	95	5	1.0
PNVCL 90-IA 10	90	10	1.0

**Table 2 gels-05-00041-t002:** UV-spectroscopy reading of phase transition of PNVCL based samples.

Polymer Code	LCST (°C)
PNVCL 100	31.8
PNVCL 95	29.8
PNVCL 90	28.4

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
