# Peer review of "Synthesis and Characterisation of Novel Temperature and pH Sensitive Physically Cross-Linked Poly(N-vinylcaprolactam-co-itaconic Acid) Hydrogels for Drug Delivery"

_gels, 2019, doi:10.3390/gels5030041_

Round 1
Reviewer 1 Report
In this study, one kind of physically crosslinked hydrogels comprising of polymers synthesized from the thermo-sensitive monomer N-vinylcaprolactam (NVCL) and pH-sensitive comonomer itaconic acid (IA) without the presence of solvent under ultraviolet (UV) light. Fourier transform infrared spectroscopy, and differential scanning calorimetry demonstrated the successful synthesis of the copolymer. Phase transition temperature was detected to investigate the influence of IA on the lower critical solution temperatures of poly(N-vinylcaprolactam) (PNVCL). In order to examine the pH and temperature sensitivity of P(NVCL-IA) hydrogels, swelling studies were carried out. In addition, they evaluate the potential use of dual sensitive P(NVCL-IA) hydrogels as controlled drug delivery systems. Some issues existing should be addressed before publication.
1. The relevant background or the research significance should be introduced at the beginning of the Abstract.
2. The rheological property of P(NVCL-IA) hydrogels should be tested to indicate their mechanical properties.
3. Given that the dual-sensitive hydrogels would be employed as drug delivery platforms in vivo, it is necessary to carry out cytotoxicity tests to illustrate their biocompatibility.
4. The logic of the Introduction part is disordered, thus makes readers confused. In the second paragraph, the authors mentioned the sensitivities of hydrogels and then turned to the crosslinking forms of hydrogels. Without any transition sentences, the sensitivities of hydrogels is described continuously.
5. Most of the references are outdated; the works published in the near three years should be cited and discussed, for example, Biomacromolecules 2019, 20 (4), 1478-1492; Materials Science and Engineering: C 2018, 82, 25-28; Advanced Science 2018, 5 (5), 1700527. Moreover, the format of references should be consistent.
6. Some minor mistakes are existing in the article. The authors should revise the manuscript carefully. Some mistakes are listed as follow.
6.1. All the abbreviations should be defined when they appear in the Abstract for the first time. For example, “UV” and “ATR-FTIR” mentioned in the Abstract did not offer corresponding full names.
6.2. In Figure 1, the significant ticks corresponding to tick labels should be marked. Besides, the absent title of abscissa axis needs to be added.
6.3. In Figure 4, the percent signs should be put over “Swelling” by another title.
6.4. In Line 80 of Page 2, “solvents” should be “solvent”.
Reviewer 2 Report
The manuscript by Fallon et al reports on synthesis and characterization of temperature and pH sensitive Poly (N-vinylcaprolactam-co-itaconic acid) (PVCL-co-IA) hydrogels to be used for drug delivery. The authors performed chemical and thermal analyses including ATR-FTIR, UV, and DSC as well as selling and drug release studies. These characterization methods are routinely used for materials characterization.
PVCL-based materials are interesting from both fundamental and applications aspects, yet, I have two main concerns about this work.
My biggest concern is about the hydrogel itself. The authors claim that this is a physically cross-linked network of PVCL and IA, but they do UV-based free-radical polymerization that should give a chemically crosslinked network. To prove that this is physically-cross-linked network of PVCL-co-IA, GPC results should be presented at basic pH when PVCL and IA do not form hydrogen bonding.
My other concern is the lack of novelty and significance. There are several articles already published on chemical and physical networks of PVCL-IA based hydrogels presenting similar characterization methods. The authors claim they do not use any solvents in the current method as opposed to others, but it is not clear how properties of the produced material are better than those published before.
Finally, I suggest deepening discussion by comparing synthesis and properties of the presented hydrogels to previously published related networks to emphasize the significance of this work.
Round 2
Reviewer 1 Report
Accept as is.